# Common anti-cancer therapies induce somatic mutations in stem cells of healthy tissue

Ewart Kuijk [1,2], Onno Kranenburg [3,4], Edwin Cuppen [2,5] ✉ & Arne Van Hoeck [2] ✉

Genome-wide mutation analyses have revealed that specific anti-cancer drugs are highly mutagenic to cancer cells, but the mutational impact of anti-cancer therapies on normal cells is not known. Here, we examine genome-wide somatic mutation patterns in 42 healthy adult stem cells (ASCs) of the colon or the liver from 14 cancer patients (mean of 3.2 ASC per donor) that received systemic chemotherapy and/or local radiotherapy. The platinum-based chemo-drug Oxaliplatin induces on average $535 \pm 260$ mutations in colon ASC, while 5-FU shows a complete mutagenic absence in most, but not all colon ASCs. In contrast with the colon, normal liver ASCs escape mutagenesis from systemic treatment with Oxaliplatin and 5-FU. Thus, while chemotherapies are highly effective at killing cancer cells, their systemic use also increases the mutational burden of long-lived normal stem cells responsible for tissue renewal thereby increasing the risk for developing second cancers.

Advances in cancer treatment are continuously increasing the lifespan of cancer patients[1,2]. Given the growing population of cancer survivors, there is a need to better understand the long-term side effects of anti-cancer therapies, including the risk of developing secondary malignancies. Systematic analysis of cancer genomes has recently revealed that both radiation[3,4] and chemotherapy[5–7] induce mutations in the tumor cells of treated cancer patients. In contrast to localized radiation therapy, chemotherapies are administered systemically, either orally or intravenously, and thus constitute a potential genotoxic risk for every non-malignant cell of the human body. The mutational impact of anti-cancer therapies on healthy adult stem cells (ASCs) is of great interest as ASCs are long-lived and responsible for tissue renewal and are considered the cells-of-origin in cancer[8,9].

Among standard of care chemotherapeutics, particularly 5-FU and platinum-based drugs induce many characteristic mutations in treatment surviving tumor cells. Computational models on in vivo mutation data from 5-FU and platinum-treated metastatic cancers suggest a mutation rate in surviving cancer cells that is respectively 50- to 100-fold higher than normal aging mutational processes[5,6]. However, the observed treatment-induced mutational loads from tumor-specific studies do not necessarily translate to healthy cells. Cancer cells are typically genomically unstable, highly proliferative, and may have selected treatment resistance mechanisms. Moreover, the treatment mutation rates were obtained from bulk whole-genome sequencing (WGS) data in cancer cells following chemotherapy. Hence, only mutations in surviving cancer cells that have undergone clonal expansion during, or after, chemotherapy treatment can be identified and quantified using this approach. Recently, clonal expansion in non-cancerous tissue has been exploited to study chemotherapy-related mutations in bulk[10,11] and in single[12] hematopoietic cells. For solid tissue, clonal mutations can be identified with bulk sequencing of microscale sampled tissue[13–15]. However, mutations of the most recent mutagenic processes that have only been active in the period just before tissue collection, such as treatment-induced mutagenesis, remain subclonal and thus undetected by this approach. Therefore, dedicated approaches that are not dependent on the degree of tissue

[1]Division of Pediatric Gastroenterology, Wilhelmina Children's Hospital, University Medical Center Utrecht, Utrecht, The Netherlands. [2]Center for Molecular Medicine and Oncode Institute, University Medical Center Utrecht, Utrecht, The Netherlands. [3]Laboratory Translational Oncology, Division of Imaging and Cancer, University Medical Center Utrecht, Utrecht, The Netherlands. [4]Utrecht Platform for Organoid Technology, Utrecht University, Utrecht, The Netherlands. [5]Hartwig Medical Foundation, Amsterdam, The Netherlands. ✉e-mail: E.cuppen@hartwigmedicalfoundation.nl; A.vanhoeck@umcutrecht.nl

clonality are essential to comprehensively determine the mutational impact of anti-cancer treatments on normal cells. While single-cell techniques are rapidly maturing[16], their genome-wide sensitivity and specificity is still limited with noise often still exceeding (very) low mutation rates in specific systems and additional challenges for detecting structural and small copy number variants.

We have developed a genome-wide approach to identify all somatic mutations acquired in vivo in single ASCs by combining clonal expansion of individual ASCs via organoid technology with WGS of the expanded clones[17,18]. Importantly, this approach is highly sensitive (<50 mutations per cell) and enables the detection of recently acquired somatic mutations[19], which would remain undetected by bulk tissue sequencing.

In this work, we applied this single-cell-based approach to identify genome-wide somatic mutations in healthy colon and liver ASCs derived from colon cancer patients that received Capecitabine/Oxaliplatin (CapOx) treatment. This regimen includes the 5-FU prodrug CAPecitabine and the platinum-based OXaliplatin. Currently, these are the most commonly used first-line chemotherapies for the treatment of many solid cancers including colon cancer. Because both drugs are administered simultaneously to the cancer patient, we can directly compare the impact of platinum and 5-FU mutagenesis in vivo in the same ASCs (i.e., genetically uniform with equal proliferation dynamics)

as these drugs leave distinct mutational footprints[5,6]. We studied mutations in ASCs derived from the slowly renewing liver (cultured as intrahepatic cholangiocyte organoids[20]) and the actively proliferating colon because these tissues show distinct mutational signatures upon aging, which suggests differences in the balance between DNA damage and repair that may affect treatment-induced mutagenesis as well[18,21]. Overall, our results indicate a differential permissiveness of ASCs in different tissues to acquire therapy-induced mutations for different chemotherapeutics.

## Results

### The in vivo mutational impact of CapOx in healthy colorectal and liver ASCs

Morphologically normal colorectal or liver tissue was collected from 14 colorectal cancer (CRC) patients who underwent CapOx treatment. Healthy colorectal organoid cultures were established of 7 CRC patients who received neoadjuvant CapOx chemotherapy with on average 3.2 (±1.7 SD) treatment cycles and a time span of 1.5 (±0.8 SD) months between the latest CapOx cycle and collecting the biopsy (Fig. 1a and Supplementary Table 1). Of these, 1 CRC patient received only 5-FU and 3 patients also received local radiation therapy. Healthy liver cholangiocyte organoid cultures were established from 7 CRC patients with liver metastases who had been treated with on average

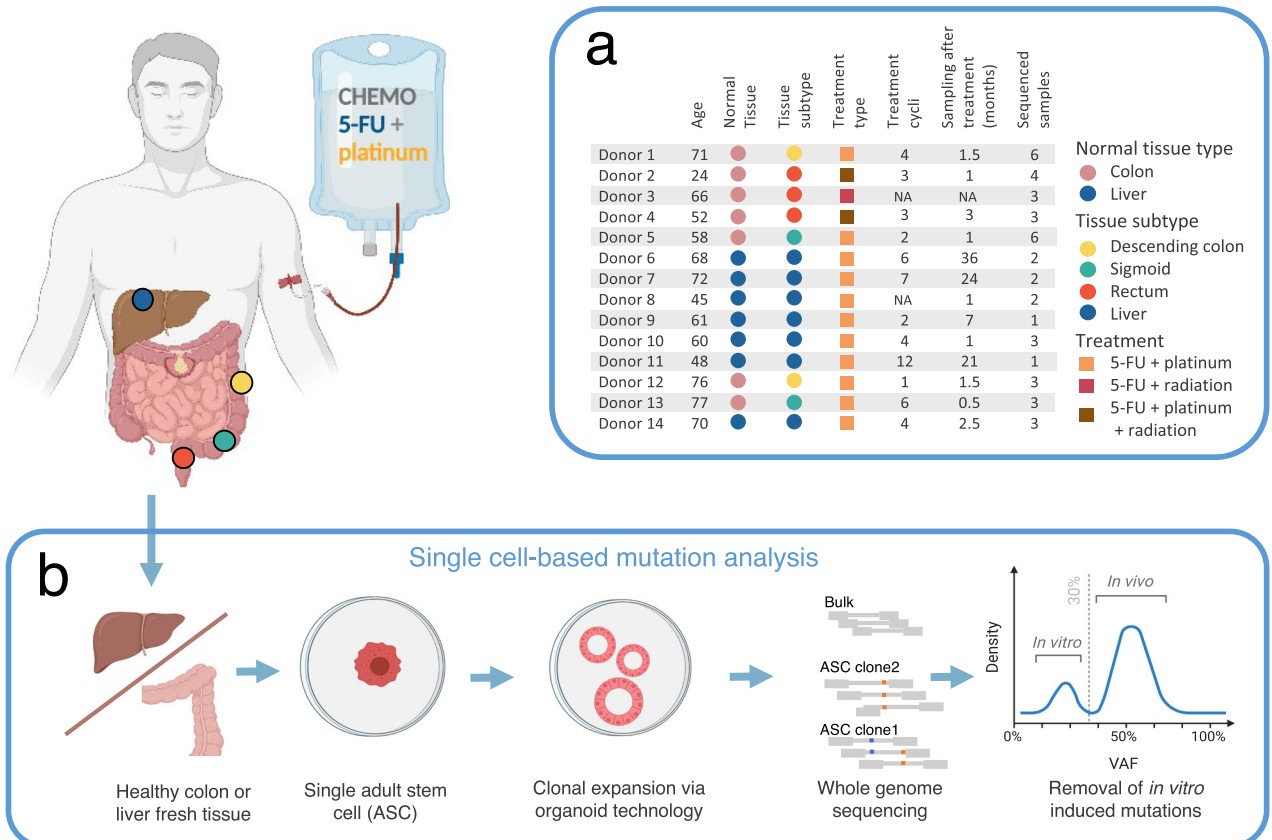

**Fig. 1 | Experimental design. a** Clinical overview of the 14 treated donors used in this study. The table describes the pathological details, treatment history and the number of adult stem cell (ASC) samples analyzed per donor. **b** Schematic of the experimental setup to determine genome-wide somatic mutations in individual healthy colorectal and liver ASC from patients who have received CapOx chemotherapy and/or radiotherapy treatment. For this, fresh colorectal and liver normal tissue was derived from the resection margin around the colorectal tumor during surgical removal of respectively primary colorectal tumors or colorectal tumors metastasized to the liver to derive healthy colon or liver tissue, respectively. Subsequently, fresh healthy tissue was minced, dissociated into single adult stem cells (ASCs) solutions and in vitro expanded into clonal organoid cultures to obtain sufficient DNA for WGS. Polyclonal tissue was also sequenced to identify and exclude germline variants. Heterozygous mutations present in the individual ASC at the start of culture display a variant allele frequency (VAF) of ~50%. Mutations that are introduced during in vitro culturing, after the single-cell step, have a VAF under 30% and were discarded. The obtained in vivo mutation dataset from single colorectal and liver ASCs from treated cancer patients were compared to mutation burdens from untreated donors[18] as well as subjected to mutational signature analysis to quantify the 5-FU (prodrug of CAPecitabine), platinum (OXaliplatin) and radiation-induced mutational impact. This figure was partly created with BioRender.com.

5.8 (±3.5 SD) CapOx treatment cycles, biopsied 15.1 (±14.4 SD) months after treatment. Of these, 1 CRC patient received 5-FU chemoradiation without Oxaliplatin. The clonal step was performed by stringent mechanical fragmentation of primary organoid cultures, followed by expansion of individually picked organoids (see methods). In total, 42 organoid clones (28 from colon tissue with a mean of 4 clones per donor and 14 from liver tissue with a mean of 2 per donor) were expanded in vitro until there was enough material available for WGS (Fig. 1b). Additionally, we obtained and sequenced a polyclonal multi-germ layer tissue sample from each patient to identify and exclude germline mutations. Analysis of the somatic variants showed variant allele frequency (VAF) distribution peaks at 50% in each ASC, confirming the clonality of the cultures[17] (Supplementary Fig. 1). Mutations acquired during culturing were excluded based on their lower VAF. To be able to control for age-related mutation accumulation, we combined the in vivo chemotherapy-treated healthy ASCs mutation dataset with a previously published dataset of normal untreated liver and colorectal ASCs, which was generated using the same experimental setup[18].

We found that 4 CapOx-treated colorectal donors had a higher single base substitution (SBS) mutation burden in individual ASCs than expected based on their age (Fig. 2a). The increase in the 24- and 52-year-old donor reached significance with a respective excess of ~1680 and 1650 mutations per cell ($p < 0.01$; linear mixed model (LMM)), and in the 71- and 77-year-old patient ~1100, and ~1020 more mutations than expected ($p = 0.024$ and $0.047$). The SBS mutation burden of the 58- and 76-year-old CapOx-treated donors, as well as the 5-FU-only treated colorectal donor, was within the same range as healthy untreated controls. In addition to SBS, the 4 colorectal donors with increased SBS mutation burden and the 76-year-old donor also harbored an increased doublet base substitution (DBS) burden (Fig. 2c). The insertion and deletion (indel) and structural variant (SV) burdens were both significantly increased in the 24- and 66-year-old donors (Fig. 2e, g). In contrast to colorectal ASCs, none of the liver ASCs showed a significantly elevated mutation burdens for any of the mutation types (with the exception for the DBS burden in the 68-year-old liver donor) (Fig. 2d). The liver patients included in this study have undergone more CAPOX treatment cycles than the colon patients and thus the lack of treatment-related mutations in liver ASCs cannot be explained by a lower number of treatment cycles (Supplementary Table 1).

## Characteristics of CapOx-induced somatic mutations

To further assess the nature of therapy-induced mutations in an unbiased way, we performed tissue-specific mutational signature analysis using two independent non-negative matrix factorization (NMF) approaches, Mutational Patterns[22] and SigProfiler[23]. For this, we appended our data set with public healthy untreated liver and colon tissue mutation datasets to allow for comparison with previously reported signatures in these organs[14,15]. Compared to the mutational signatures from normal untreated colon[15], NMF identified one distinct de novo SBS and one distinct de novo DBS signature, which were only present in the CapOx exposed colorectal ASCs (Fig. 3a, b, d, e and Supplementary Figs. 2–6). Because CapOx consists of two distinct chemotherapy drugs that are administered simultaneously, we reasoned that this distinct NMF extracted SBS signature could represent the concurrent activity of two distinct mutational processes: a platinum-based process and a 5-FU-based process. Indeed, this de novo SBS signature showed a strong resemblance to a combination of the COSMIC[24] 5-FU (SBS-17) and platinum (SBS-35) signatures (cosine sim = 0.88) (Fig. 3a, b). Similarly, the distinct de novo DBS signature resembles the COSMIC platinum DBS (i.e., DBS-5) signature and its mutation contribution is also increased in CapOx-treated colorectal ASCs compared to untreated colorectal ASCs (Fig. 3d, e). In the extracted mutational patterns from the liver samples, no additional

signatures from previous reported mutational signatures of untreated liver tissue were identified[14], nor was there a difference in the activity of mutational processes observed in CapOx-treated liver ASCs compared to untreated liver ASCs (Supplementary Figs. 7–9). Even the highly characteristic CT > AA and CT > AC mutations, which are strongly associated with platinum mutagenesis[23], were not found enriched in treated liver ASCs (Supplementary Fig. 10). Because the time between treatment and collection was on average several months longer for the liver than for the colon, damaged cells may have been effectively cleared from the liver. However, also no treatment-related mutations were observed in ASCs of the two liver donors that were collected over a short time span of 1 month after treatment. Therefore, these results indicate that CapOx chemotherapy is mutagenic in vivo in normal colorectal ASCs, whereas normal liver ASCs appear to effectively escape 5-FU and platinum drug-induced mutagenesis.

To separately quantify 5-FU and platinum mutations in colorectal ASCs, we fitted the mutational data to the well-established COSMIC mutational signatures[24]. As expected, the age-related SBS mutation load (mutational signatures SBS-1, 5 and 18[15,18]) increased linearly with donor age (Supplementary Fig. 11a, b). The annual mutation rate for these signatures was similar for both untreated controls (43 mutations per year; LMM; 95% CI: 27–60) and CapOx-treated colorectal ASCs (43 mutations per year; LMM; 95% CI: 31–55) (Supplementary Fig. 11c), indicating that the overall increase of the SBS burden observed in four colorectal donors cannot be explained by acceleration of age-related mutational processes.

Regarding chemotherapy-induced mutations, we found that each ASC of five out of six CapOx-treated colorectal donors showed a platinum-associated SBS-35 mutational load that varied between 500 to 1000 mutations (Fig. 3c). The other CapOx-treated donor, aged 76, showed 445 SBS-35 mutations in 1 ASCs, while no SBS-35 mutations were detected in the 2 other ASCs of this donor. Reassuringly, the colorectal donor that did not receive platinum treatment lacked any SBS-35 contribution. DBS-5 mutational load varied between 5–50 mutations (Fig. 3c) and was strongly correlated with SBS-35 mutational load, which reflects an equal SBS and DBS platinum mutation rate of 1 DBS mutation per 26 (±2 SE) SBS mutations (Fig. 3f) (linear model, Pearson's $r = 0.88$). Other DBS mutation types, of which some overlap with age-related DBS contexts, were also increased in platinum-treated colorectal ASCs (Supplementary Fig. 12). This indicates that platinum may induce mutations in more DBS contexts than CT > AA and CT > AT mutations. This is consistent with in vitro DBS context data in platinum-treated induced pluripotent stem cells[25].

Next, we found that the number of subsequent CapOx treatment cycles is an important determinant for the platinum mutational effect. Each CapOx cycle induced an additional 105 (±33 SE) SBS and 5 (±1 SE) DBS mutations (t-test LMM; $p_{SBS} = 0.03$; $p_{DBS} = 0.03$; Fig. 3g and Supplementary Fig. 13). The two CapOx-treated donors with normal mutational burden received the lowest number of treatment cycles (i.e., 1 and 2 cycles) which in part explains why they display a mutation burden within the range expected for untreated donors. This may also indicate that Oxaliplatin was mutagenic in every ASCs of the 76-year-old colon donor that received only a single CAPOX cycle, but its contribution to the total burden in 2 ASCs remained under the detection limit of mutational signature analysis.

The consistent platinum mutational impact across donors, as well as the limited variation of platinum-induced mutations between different cells from the same donors, indicates that each platinum treatment results in predictable additional mutational load in any healthy colorectal ASCs equivalent to 10–20 years of normal age-related mutagenesis. In line with this, a similar platinum-induced mutational load (430 SBSs) was previously reported in colon crypts of a single patient treated with cisplatin[26]. With an estimated $10^8$ colorectal stem cells[27] in the adult colon, each platinum treatment leads to an enormous pool (>$10^{10}$) of potentially deleterious mutations that

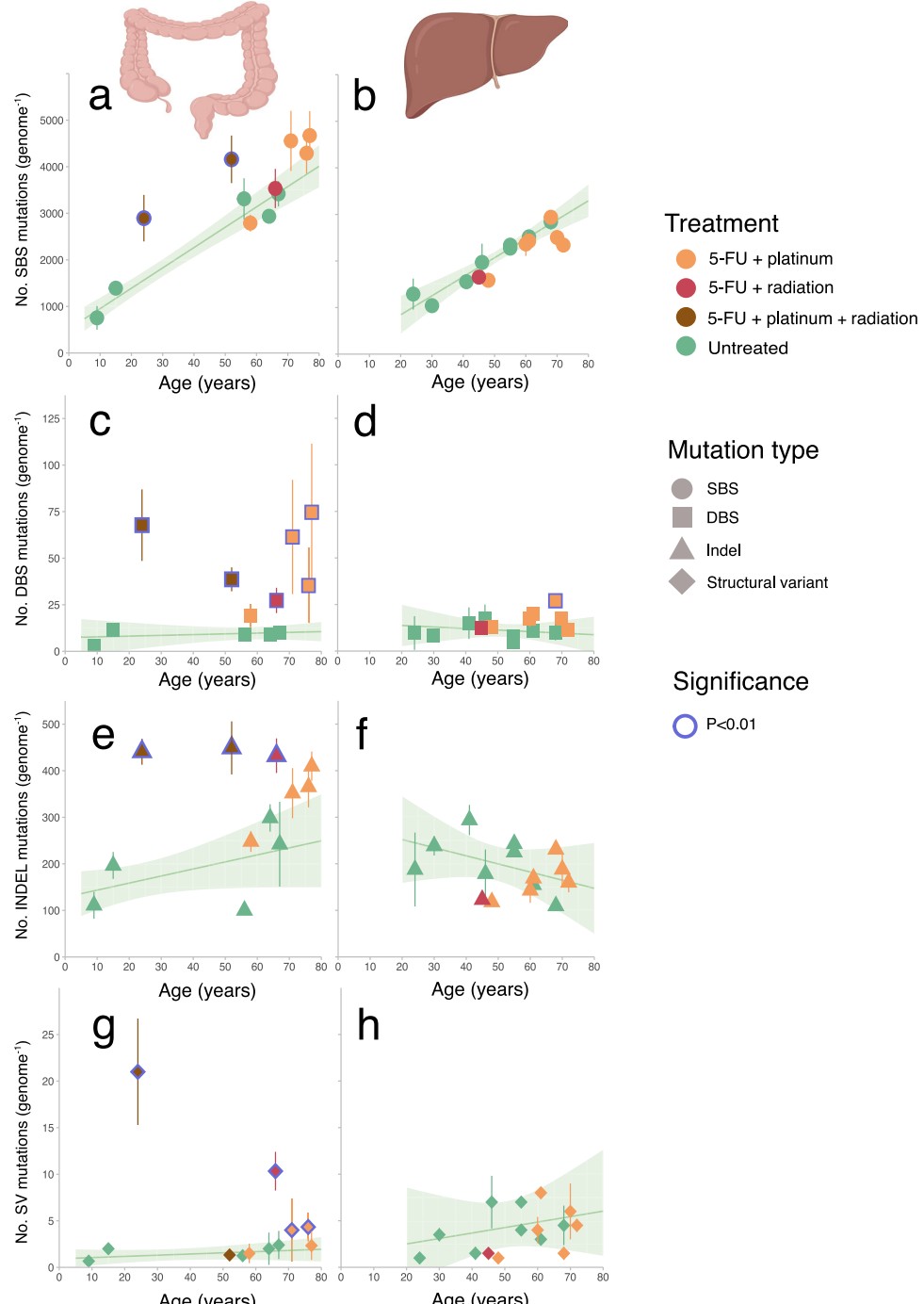

**Fig. 2 | Mutational impact on mutation burden. a, b** Singe base substitution (SBS), **c, d** double base substitution (DBS), **e, f** indel and **g, h** structural variant (SV) mutation burden (*y* axis) as a function of age (*x* axis) for respectively healthy colorectal and liver ASCs. Each data point represents the mean mutation burden per donor and the error bars represent the standard deviation of the mutation burden. The color of each point depicts the treatment history. The number of sequenced ASC per donor (*n*) varies from 1 up to 6 samples and is listed in Fig. 1a. The cohort size for treated and untreated colorectal ASCs are respectively 6 and 7 donors while 8 and 7 donors are included for respectively the untreated and treated liver cohort. The green lines display the expected mutation burden of the indicated mutation types calculated from untreated ASCs using a bootstrapped linear mixed effects model (LMM) approach. The shaded areas cover the standard deviation of the LMM of the corresponding regression lines. Treated donors with a significantly increased mutation burden (i.e., more than expected from normal aging modeled from untreated donors ($p < 0.01$)) are marked with blue circles. This figure was partly created with BioRender.com. Source data are provided as a Source Data file.

may explain why platinum drugs pose an increased risk for the development of secondary malignancies[28].

Regarding 5-FU mutagenesis, readily detectable by COSMIC SBS-17[5], we found much more heterogeneity in mutation contribution between the seven 5-FU-treated colorectal donors. The 24-year-old

donor accumulated on average 265 (±195 SD) 5-FU-induced SBS mutations in every surveyed colorectal ASC (Fig. 3c). In contrast, a complete absence of the 5-FU-related footprint was observed in every analyzed colorectal ASC for five out of seven 5-FU-treated donors. Intriguingly, the 66-year-old donor acquired 282 5-FU-induced

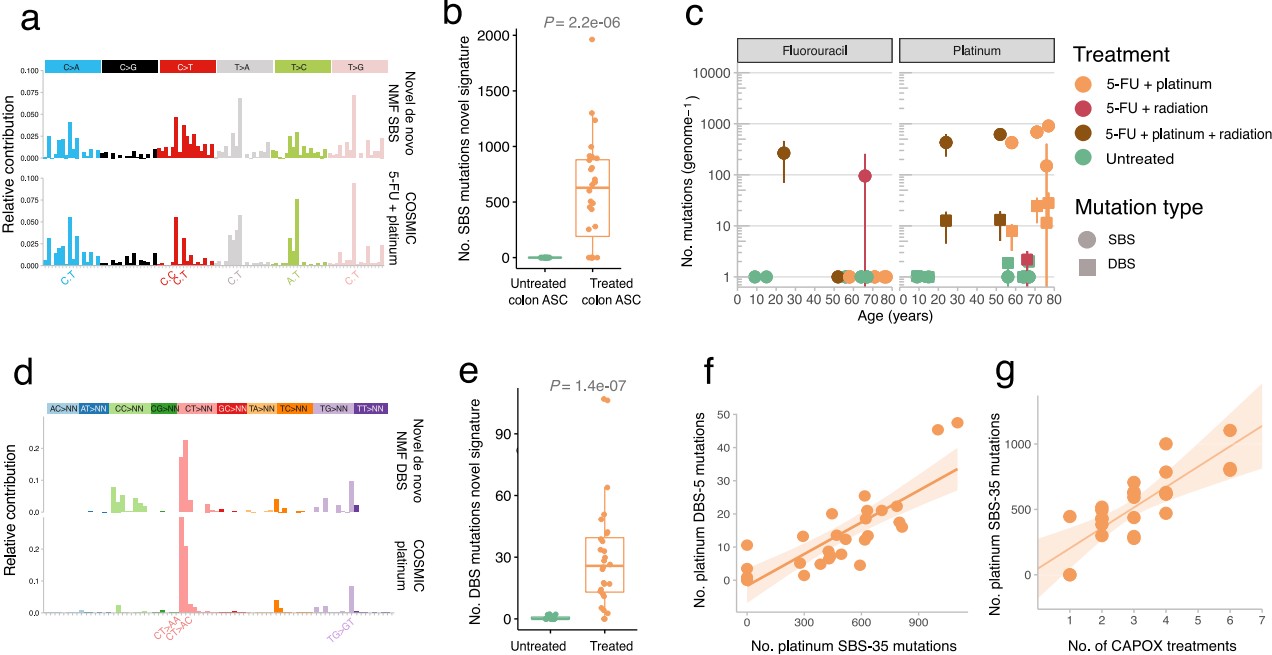

**Fig. 3 | Chemotherapy-induced mutations in healthy colorectal ASCs. a** The distinct extracted de novo SBS signature in treated healthy colorectal tissue resembles a mix of the COSMIC 5-FU and platinum mutation signatures (cosine sim = 0.88). **b** Box and whisker plot indicating the relative mutation contribution of the treated specific SBS mutation signature from colorectal ASCs between CapOx-treated ($n = 7$) and untreated ($n = 5$) colorectal donors. The here shown box and whiskers plot displays the first and the third quartiles (top and bottom of the box), the median (vertical line inside the box), the extremes (whiskers) and the single data points (single dots). A Wilcoxon rank-sum test between every cohort was performed and the $p$ value is illustrated at the top of the plot. **c** 5-FU and platinum SBS and DBS mutation contributions ($y$ axis) for each donor as a function of age ($x$ axis). Each data point represents the mean mutation contribution per donor, the error bars represent the standard deviation and the color depicts the treatment history. The number of sequenced ASC per donor ($n$) varies from 1 up to 6 samples and is listed in Fig. 1a. The 5-FU and platinum mutation contributions are derived with refitting on the well-established platinum COSMIC signatures. **d** The distinct

extracted de novo DBS signature in CapOx-treated ASCs from ($n = 7$) healthy colorectal donors. This signature resembles the COSMIC platinum DBS-5 mutation signature (cosine sim = 0.81). **e** Box and whisker plot indicating the relative mutation contribution of the treated specific DBS mutation signature from colorectal ASCs between oxaliplatin-treated ($n = 7$) and untreated ($n = 5$) donors. A Wilcoxon rank-sum test between every cohort was performed and the $p$ value is illustrated at the top of the plot. **f** Scatterplot showing the relation between platinum SBS ($x$ axis) and DBS ($y$ axis) mutations, derived with refitting on the well-established platinum COSMIC signatures. The brown line displays the least-squares linear fit of ($n = 25$) oxaliplatin-treated ASCs while the shaded region represents the 95% confidence interval of the fit (Pearson's $r = 0.88$). **g** Scatterplot showing the relation between the number of CapOx treatments ($x$ axis) and platinum SBS mutations ($y$ axis) of the ($n = 25$) oxaliplatin-treated ASCs. The $p$ value of the treatment count effect in the LMM that controls for donor ID (two-tailed $t$-test) is 0.03. The shaded area represents the standard deviation of the LMM by treatment count effect. Source data are provided as a Source Data file.

mutations in one ASC, whereas the other ASC from this donor completely lacked 5-FU-induced mutations. This heterogeneous 5-FU mutation landscape in healthy colorectal ASCs may be related to pharmacodynamic variation between donors, differences in dosing, or different treatment schedules. However, this does not explain the observed intra-donor heterogeneity in the 66-year-old donor as this involves isogenic normal cells that have been exposed to the same treatment regimen. These findings may indicate that 5-FU mutagenesis depends on specific proliferative or metabolic conditions in target cells.

In eight out of twenty-five colorectal ASCs, we observed contributions of platinum-induced SBS and DBS mutations in low VAF ranges reminiscent of subclonal mutations accumulated during culturing (Supplementary Fig. 14a). However, the culture medium did not contain any substances known to induce these specific mutations. Maybe the individual colorectal organoids of these clones represent two stem cells of the same crypt that originate from a common progenitor stem cell during treatment, but then a similar subclonal contribution would be expected for 5-FU-induced mutations (at least for the 24-year-old donor in whose ASCs both 5-FU and platinum mutations have accumulated (Supplementary Fig. 14a)), which was not the case. Alternatively, cytotoxic platinum compounds may have remained present as DNA adducts for a longer period and were still present at the moment of tissue harvesting. Such mutagenic adducts

could subsequently segregate during in vitro expansion resulting in subclonal platinum mutations. This lesion segregation phenomenon has recently been described for many carcinogens, including platinum-based compounds[29]. Such DNA complexes generate low-VAF mutations when some lesions are properly repaired in the daughter cells. 5-FU-induced mutations are not caused by DNA complexes that segregate over multiple cell cycles, which may explain why subclonal SBS-17 mutations were detected with a considerably lower mutation load than platinum mutations (Supplementary Fig. 14a). Although it remains unclear which mechanism, or combinations thereof, explains the subclonal platinum-related mutations, it is important to note that the clonal platinum mutation loads we report may thus be an underestimation of the total platinum mutational impact for some of the assessed single colorectal ASC. Nevertheless, only two ASCs (from the 24-year-old donor) showed a subclonal platinum mutation contribution of 100% or more and the inclusion of subclonal platinum mutations had a low impact (less than 50%) on the platinum mutational load for all other ($n = 25$) ASCs (Supplementary Fig. 13b, c).

## Radiotherapy induces large and structural mutations

Two out of three rectal cancer patients (24- and 66-year-old) treated with local radiation therapy harbored a significant increase in indel and SV burden (Fig. 2). De novo mutational signature analysis revealed that the colorectal ASCs from these patients showed activity of an indel

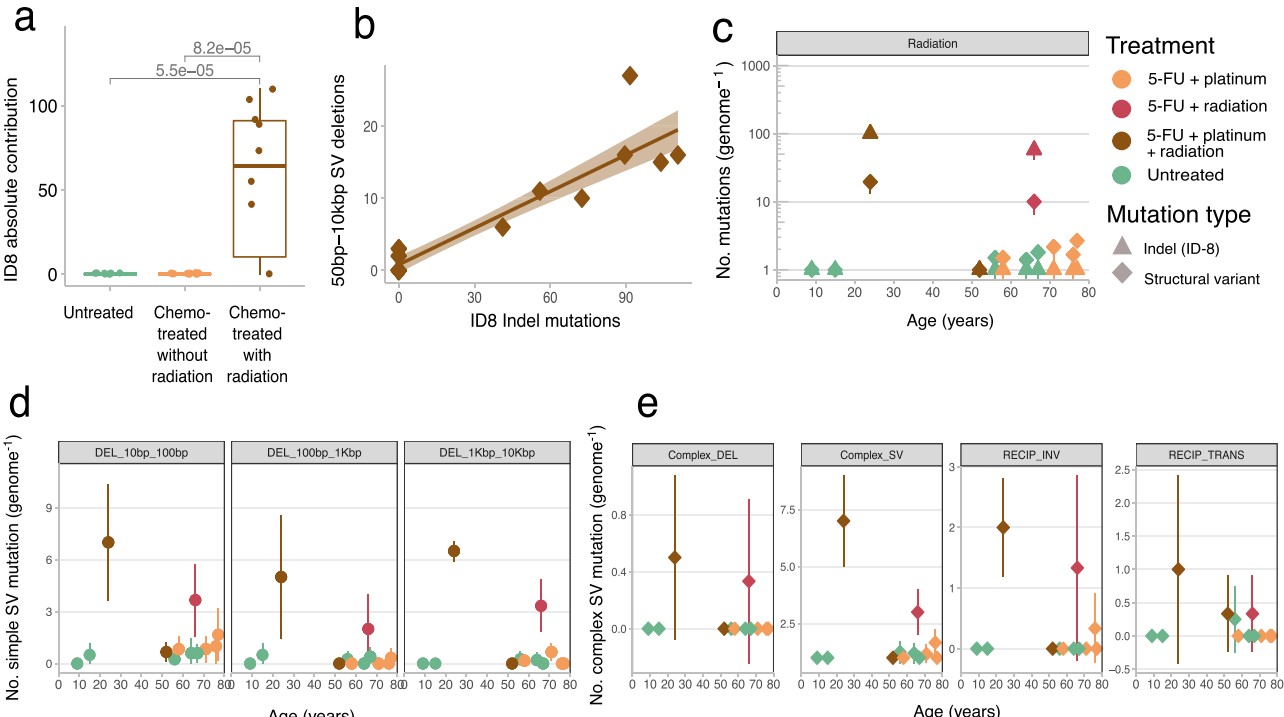

**Fig. 4 | Radiotherapy-induced mutations in healthy colorectal ASCs. a** Box and whisker plot indicating the absolute mutation contribution of COSMIC ID-8 indel signature between chemo-treated ASCs with ($n = 10$) and without ($n = 18$) radiotherapy and untreated ($n = 19$) colorectal ASCs. The box in the boxplot delimits the first and third quartiles of the distribution (with a line representing the median); the whiskers delimit the lowest data point above the first quartile minus 1.5 times the interquartile distance and the highest data point below the third quartile plus 1.5 times the interquartile distance. A Wilcoxon rank-sum test between every cohort was performed and the $p$ value is illustrated at the top of the plot. **b** Scatterplot showing the relation between platinum ID-8 mutations ($x$ axis) and structural deletion variants ($y$ axis). The brown line displays the least-squares linear fit while the shaded region represents the 95% confidence interval of the fit (Pearson's $r = 0.92$). **c** Radiotherapy-induced indel and structural variant mutation

contributions ($y$ axis) for each donor as a function of age ($x$ axis). Each data point represents the mean mutation contribution per donor for the indicated mutation type. The error bars represent the standard deviation and the color of each point depicts the treatment history. The ID-8 mutation contributions are derived with refitting on the well-established COSMIC signatures. **d, e** Each data point represents the mean structural mutation contribution per donor, the error bars represent standard deviation and the color of each point depicts the treatment history. **d** Simple structural deletions (DEL) split by length, and **e** complex structural deletions subdivided by complex SV type: Deletion (DEL), SV (structural variant), RECIP_INV (reciprocal inversion) and RECIP_TRANS (reciprocal translocation). The number of sequenced ASC per donor ($n$) varies from 1 up to 6 samples and is listed in Fig. 1a. Source data are provided as a Source Data file.

signature characterized with >5 bp small deletions without microhomology context ($p < 0.05$, Wilcoxon rank-sum test, Supplementary Fig. 15). This signature was mostly similar to COSMIC ID-8 (cos sim = 0.66). ID-8 mutations were previously found to be enriched in human cancers following radiation therapy[3,4]. Indeed, refitting the indel mutation data to the COSMIC ID signatures showed that ID-8 mutations were restricted to the ASCs of the two radiation-treated donors with 99 (±10 SD) and 66 (±16 SD) ID-8 mutations per cell (Fig. 4a). The ID-8 footprint has been linked to DNA double-strand break repair by non-homologous DNA end-joining mechanisms, which is in line with the type of DNA damage induced by radiotherapy. The colorectal ASCs of the radiotherapy-treated 52-year-old donor with normal ID burden lacked ID-8 mutations, and were likely collected from normal tissue that remained unexposed to radiotherapy.

In addition to small deletions, we investigated SVs because radiation-treated tumors show radiotherapy-associated increases of inversions and large-scale deletions[3,4]. The irradiated ASCs of the two donors also showed a remarkable SV burden of 21 (±6 SD) and 10 (±2.1 SD) SVs, of which >95% consisted of 50 bp to 10 kb deletion events (Fig. 4c, d and Supplementary Fig. 16). As with ID-8 mutations, also these larger deletions contained little or no microhomology sequences at the breakpoints and their contribution was strongly correlated with ID8 mutation load (Fig. 4b and Supplementary Fig. 17). Complex structural events were also observed in every radiation-treated ASC, including multi-breakpoint rearrangements, complex deletions, and

reciprocal events, which are generally absent in non-irradiated ASCs (Fig. 4e). The 24- and 66-year-old donors only received radiotherapy and CapOx chemotherapy (66-year-old only 5-FU), which indicates radiation exposure caused these complex SVs as they are not detected in the ASCs from patients that received only chemotherapy. Complex SV events are frequently observed as a result of positive selection, because genes are affected that can contribute to tumor development and treatment resistance. Here, we did not observe any complex SV mutations, or other mutation types, in oncogenes or tumor suppressor genes (e.g., *TP53*, which is important for genomic instability), indicating the observed events are likely all passenger mutations that do not confer a selective advantage. Nevertheless, the simple and complex structural rearrangements in normal ASCs demonstrate that radiation therapy can cause severe structural DNA damage without necessarily compromising cell viability, as these cells were readily cultured in vitro, and may explain why solid second cancers associated with radiotherapy occur in or near the area that was irradiated[30].

## Discussion

Our findings reveal that both CapOx chemotherapy and radiotherapy are mutagenic in colorectal ASCs and significantly increase the mutational burden in normal non-cancerous cells beyond typical age-related mutation accumulation. Unlike colorectal ASCs, liver ASCs do not acquire such somatic mutations resulting from systematic CapOx treatment. We also found that the CapOx mutagenic effect in

colorectal ASCs is chemo-drug dependent. The platinum-induced mutational load was highly consistent within individuals and between donors. In contrast, 5-FU treatment resulted in completely different mutation contributions, with considerable inter- and intra-individual variation. In fact, most colorectal ASCs showed a complete absence of 5-FU-induced mutations. On average, there was a ~13 month longer time span between the end of the CapOx treatment regime and tissue sampling for liver donors than for colon donors, which may explain the absence of chemotherapy-related mutations in liver ASCs. However, healthy tissue of 2 liver donors (donor 8, 10) was collected 1 month after the end of CapOx treatment, which is below the average of 1.5 month for colon donors and all 5 liver ASCs of these 2 liver donors showed a complete absence of CapOx-induced mutations as well. This indicates that time between treatment and biopsy is likely not the cause for the lack of CapOx related mutations for liver ASCs. The lack of treatment-related mutations in liver ASCs can also not be explained by a lower number of treatment cycles, because the liver tissues have been exposed to more CapOx treatments than the colon tissues. And although we cannot formally exclude the possibility of a highly efficient clearance of treatment damaged healthy liver ASCs, we consider this hypothesis unlikely because 5-FU or platinum therapies rarely induce liver damage[31]. The pyrimidine analog 5-FU is thought to impair the nucleotide pool (dNTPs) that is used for DNA synthesis[32]. Therefore, active DNA replication and cell proliferation are likely essential for 5-FU mutagenesis because damaged nucleotide precursors can be amended by DNA repair mechanisms and nucleotide salvage pathways in non-dividing cells. A large inter-individual variation was also observed for the purine signature in colorectal crypts from inflammatory bowel disease patients[26], indicating that mutation contributions from nucleotide precursors in general, are highly heterogeneous. In contrast, platinum may be less dependent on the cell cycle because platinum complexes form DNA cross-links through covalent bonds, which does not require a specific proliferative state at moment of administration to be mutagenic, but requires error-prone repair to be fixed as a mutation in later cell division(s). The different underlying mutational mechanisms between 5-FU and platinum-based drugs may underlie the large inter- and intra-individual variability in mutation contribution for 5-FU and the low variability for platinum drugs.

Although the absence of 5-FU-induced mutations in slowly proliferating liver ASCs could be anticipated based on its mode of action, the complete lack of platinum mutations was surprising. Liver ASCs have a comparable age-related mutation burden to colorectal ASCs[18], demonstrating that liver ASCs are not exempt from DNA damage and erroneous repair. Given the liver anatomy, it is improbable that liver cells fully escape exposure to the therapeutic drugs. Therefore, it is tempting to speculate that liver ASCs have more effective inherent mechanisms to protect against environmental mutagenic processes. This speculation is supported by the observation that liver stem cells from alcoholic, primary sclerosing cholangitis, and nonalcoholic steatohepatitis patients do not have increased mutational loads compared to ASCs from healthy individuals[33]. However, sporadic mutations from exogenous factors, such as smoking and aristolochic acid, have been found in normal liver tissue[14]. Alternatively, liver ASCs with a normal mutation burden may have been preferentially selected (in vivo or during in vitro expansion) over therapy-mutated liver ASCs and sequenced because of their better fitness. A similar hypothesis has been proposed for ex-smokers where, despite their smoking history, a significant fraction of normal lung cells show a normal mutation burden that may originate from physically protected quiescent stem cells that have avoided exposure to tobacco carcinogens[34]. Additional more-focused analyses (spatial and/or single-cell-based approaches) will achieve a better understanding of why healthy liver tissue is often spared from mutation accumulation as a result of systematic chemotherapy treatment.

In conclusion, our results demonstrate first-line CapOx chemotherapy imposes a tissue-specific mutational impact on healthy ASCs. Therefore, tissues with similar vulnerability as the colon may be more susceptible to developing secondary malignancies following treatment with mutagenic chemo-drugs, whereas other tissues, such as the liver, are protected from CapOx mutagenesis. To fully understand and decrease the risk of secondary malignancies, future work should aim to further elucidate the mutagenic effects of chemotherapies on other organ systems and organ anatomical sublocations as well as understanding mechanisms that protect some tissues from acquiring therapy-induced mutations.

## Methods

### Tissue biopsies and isolation and culture of adult stem cells

Tissue samples from CRC patients were obtained from University Medical Center Utrecht (UMCU) pathology department within the Biobanking protocol HUB-Cancer TCBIO, which was approved by the medical ethical committee of the UMCU. Written informed consent from the donors was obtained prior to acquisition of the specimen for research use in the present study. Fresh colon or rectum normal tissue was obtained from the resection margin around the colorectal tumor during surgical removal while fresh normal liver tissue was obtained from the resection margin around the malignant colorectal tumors that had been metastasized to the liver.

For the isolation of colonic crypts, tissue was collected on ice in cold Adv+++ (advanced DMEM/F-12 supplemented with 1% (vol/vol) GlutaMAX, 10 mM HEPES, and 1% (vol/vol) penicillin–streptomycin (P/S)). The tissue was washed and transferred to a 10 cm dish, cut into small pieces and the minced tissue was transferred to a 15 ml tube. The tissue was subsequently washed in complete chelating solution (CCS) composed of sterile water supplemented with 5.6 mM $Na_2HPO_4 \cdot 2H_2O$, 7.9 mM $KH_2PO_4$, 95.8 mM NaCl, 1.6 mM KCl, 43.8 mM Sucrose, 54.9 mM D-Sorbitol and 0.5 mM dithiothreitol (all purchased from Merck) until the supernatant was clear. Subsequently, the biopsies were transferred to one well of a 6-well plate containing 3 ml CCS supplemented with 120–180 µl of 0.5 M EDTA and incubated at 4 °C for 30–60 min with occasional shaking and pipetting to release the crypts. When abundant crypts were visible, the biopsies were transferred to a 15 ml tube and mixed with 5 ml CCS and 2 ml fetal bovine serum (FBS). After the tissue settled, the supernatant containing the crypts was transferred to a 15 ml tube and washed twice in Adv+++ to remove the CSS. After the final wash step, the supernatant was completely removed and the pellet was mixed with Matrigel (Corning) and plated in $4 \times 10$ µl droplets per well of a 24-well plate. After the Matrigel droplets had settled at 37 °C, 500 µl of complete human intestinal organoid medium was added to the well, consisting of Adv+++ supplemented with 20% R-Spondin I conditioned medium (produced in house) 1xB27 supplement (Fisher Scientific), 20 mM nicotinamide (Merck), 1.25 mM N-Acetylcysteine (Merck), 100 ng/ml recombinant human Noggin (Peprotech), 10 µM SB202190 (Merck), 0.5 µM A83-01 (Tocris), 50 ng/ml hEGF (Peprotech), 0.5 nM Wnt Surrogate (U-Protein Expres BV), and primocin (Invivogen). Intestinal crypts have a clonal origin caused by genetic drift[35], however the treatment may have taken place in between the in vivo clonal step and the isolation of the crypts. To isolate clonal lines that also capture these recent events, we adapted the protocol, by first establishing the culture followed by severe fragmentation and serial dilutions at the first split, where individual tiny fragments composed of few cells that represent individual cells at isolation or have a very recent common ancestor in vivo. More specifically, bulk cultures were first expanded for 1–2 weeks, after which the organoids were mechanically fragmented with a pulled Pasteur's pipette into small pieces composed of very few cells. These were subsequently plated in a limiting dilution series and individual organoids were subsequently manually picked from the lowest possible concentration and further expanded until enough material was available

for whole-genome sequencing. Clonality of the clones was confirmed by VAF.

Liver stem cells were derived from cholangiocytes and not hepatocytes. To isolate liver ASC, fresh liver tissue was collected on ice in cold Adv+++ and washed in cold DMEM (Fisher Scientific) with 1% FBS and P/S. The biopsy wash transferred to a dry petridish and minced thoroughly with two scalpel blades. The minced tissue was subsequently transferred to a 15 ml tube with 4 ml Earle's Balanced Salt Solution with Ca2+/Mg2+ (Fisher Scientific) supplemented with 1 mg/ml collagenase from Clostridium histolyticum type IA (Merck) + 0.1 mg/ml DNase I (Merck). The tissue was incubated for 30 min at 37 °C in a water bath and shaken every 5 min. The tissue was further dissociated to single cells by pipetting and subsequently transferred to a 50 ml tube. The single-cell suspension was washed with cold DMEM with 1% FBS and P/S. followed by two washes with Adv+++. The cell pellet was resuspended in Matrigel and plated as a limiting dilution series in 40 μl droplets per well of a 24-well plate. After the Matrigel droplets had settled at 37 °C, 500 μl of human liver organoid establishment medium was added composed of Adv+++ supplemented with 10% R-Spondin I conditioned medium CM 10%, B27 supplement without Vit A, 10 mM Nicotinamide, N2 supplement (Thermo Scientific), 1.25 mM N-Acetylcysteine, 5 μM A83-01, 10 μM Forskolin, 100 ng/ml human recombinant FGF-10 (Peprotech), 25 ng/ml human recombinant HGF (Peprotech), 10 nM Gastrin (Merck), 50 ng/ml hEGF 500 ng/μl, 0.3 nM Wnt Surrogate, 100 ng/ml recombinant human Noggin, 10 μM Y-27632 (Bio-Connect), hES cell cloning & recovery supplement (Tebu bio) and primocin. After 2–3 days, medium was replaced with maintenance medium composed of Adv+++ supplemented with 10% R-Spondin I conditioned medium CM 10%, B27 supplement without Vit A, 10 mM Nicotinamide, N2 supplement (Thermo Scientific), 1.25 mM N-Acetylcysteine, 5 μM A83-01, 10 μM Forskolin, 100 ng/ml human recombinant FGF-10 (Peprotech), 25 ng/ml human recombinant HGF (Peprotech), 10 nM Gastrin (Merck), 50 ng/ml hEGF 500 ng/μl, and primocin. After 10–12 days individual organoids were manually picked from the lowest possible concentration and further expanded until enough material was available for whole-genome sequencing.

### Sequencing and data analysis

DNA was isolated from cell pellets with the Qiasymphony (Qiagen) DNA isolation method and the Illumina TruSeq Nano DNA Library Prep Kit was used for library preparation. Samples were sequenced on HiSeq Xten or NovaSeq6000 platforms (Illumina) with 30x coverage at Hartwig Medical Foundation sequencing services. All samples were analyzed with the HMF pipeline V4.8 (https://github.com/hartwigmedical/pipeline) which was locally deployed using GNU Guix with the recipe from https://github.com/UMCUGenetics/guix-additions. Full pipeline description is explained in ref. 36, and details and settings of all the tools can be found at their Github page. Briefly, sequence reads were mapped against human reference genome GRCh37 using Burrows-Wheeler Alignment (BWA-MEM) v0.7.5a[37]. Subsequently, somatic SBSs, double base substitutions (DBSs) and small insertions and deletions (INDELS) were determined by Strelka v1.0.14[38] that are further annotated by PURPLE. PURPLE (v2.53) combines B-allele frequency (BAF) from AMBER (v3.3), read depth ratios from COBALT (v1.7), and SVs from GRIDSS[39] to estimate copy number profiles, VAF, variant clonality and microhomology context at the breakpoints. To obtain high-quality somatic mutations that can be attributed to in vivo mutagenesis in the ASC clones, we only considered somatic mutations with a PURPLE derived VAF higher than 30% as mutations that fall outside this range were potentially induced in vitro after the clonal passage.

Analysis of the SVs was based on the LINX (v1.26)[40] output which interprets and annotates simple and complex SV events from PURPLE and GRIDSS output. LINX chains individual SVs into SV clusters and classifies these clusters into various event types. Clusters can have one SV (for simple events such as deletions and duplications which all have 1 cluster Id), or multiple SVs, with ClusterId>1 and here considered as complex SV. We defined SV load as the total number of simple SV events. We quantified deletions and duplications (ResolvedType is "DEL" or "DUP") stratified by length (1–10 kb, 10–100 kb, 100 kb–1 Mb, 1–10 Mb, >10 Mb). For complex SVs, we included "Complex_SV", |" Complex_DEL", "RECIP_INV" and "RECIP_TRANS" under resolved_Type annotation feature.

### Mutation burden analysis

The SBS, DBS and indel mutations from treated and untreated colorectal and liver ASCs were parsed from PURPLE vcfs by our developed R package Mutational Patterns[22] that was recently updated with DBS and indel functionality as well as COSMIC compatibility[41]. For each mutation type, we defined mutation burden as the total number of mutations of the autosomal genome. To statistically assess the mutation impact on mutation burden, we corrected for age-induced mutations because age is an important predictor of mutation burden. To reveal the variance on mutation burden at varying age, we performed a bootstrap resampling approach with LMMs on mutation data from untreated samples from the nlme R package[42]. LMM models allow to assess the relationship between the effect of age on the mutation burden, while correcting for multiple measurements of the same donor. We performed this procedure 10,000 times by randomly subsampling 70% of all untreated samples and computed for each bootstrap the expected mutation burden for every age interval from 5 to 80. For each mutation type and age interval, we obtained a vector of simulated distribution of the expected mutation burdens that allowed us to calculate an empirical $p$ for the mean mutation burden of each treated donor as the proportion of instances where the mean mutation burden is higher over the bootstrapped mutation burdens[43].

### Mutational signature extraction

Non-negative matrix factorization (NMF) is an unsupervised approach that decomposes high-dimensional datasets in a reduced number of meaningful mutational signatures. Two independent NMF approaches for de novo mutation signature extraction, Mutational Patterns[22] and SigProfiler[23] were used in this study. The number of somatic mutations falling into the 96 SBS, 78 DBS and, 83 indel contexts (as described in COSMIC: https://cancer.sanger.ac.uk/signatures/) was determined using Mutational Patterns to construct the mutational matrices for respectively SBS, DBS and indels. Mutation data from Lee-six et al. and Brunner et al. was parsed to mutation contexts compatible for Mutational Patterns use.

De novo mutational signature extraction was performed using the NMF Mutational Patterns extraction function with 100 iterations for each mutation type. The sigProfilerExtractor function from SigProfiler (v1.1.1) was used with default settings. The number selected signatures for each mutation type was chosen by comparing the de novo extracted colorectal and liver signatures with the reported signatures from Lee-six et al. and Brunner et al. using a cosine similarity score of 0.8 from the Mutational Patterns R package as a measure of closeness (except for 2 DBS signatures). The relative contribution of each de novo signature was calculated by dividing the absolute counts by the mutation burden of the sample using mutation counts from de novo signatures extracted with Mutational Patterns.

### Mutational signature contributions

To accurately quantify the 5-FU and platinum mutations, we fitted the mutations to the well-established COSMIC mutational signatures using the fit_to_signatures_strict function from Mutational Patterns with default settings. To avoid overfitting, only the well-described aging mutational signatures in healthy colorectal and liver tissue and which showed a de novo mutation contribution of 10% in any sample

were included. As a result, the aging signatures (SBS-1, SBS-5, SBS-18) together with 5-FU (SBS-17), platinum (SBS-35) and collibactin[44] (SBS-88) were selected. For the 5-FU signature, we merged SBS17a and SBS17b since we have previously shown that 5-FU induce a mutation footprint that combines both signatures[5]. COSMIC SBS-35 was selected over SBS-31, a signature that is also associated with platinum mutagenesis, since the de novo signature exhibited the highest similarity with SBS-35. For DBS, we included the DBS-2, DBS-4, DBS-6, DBS-9 and DBS-11 as aging signatures and DBS-5, the platinum DBS signature. Lastly, for the indels we considered ID-1, ID-2 and ID-5 as aging signatures and ID-8 (NHEJ DNA break repair) and ID-18 (colibactin).

### Detection of significantly mutated genes

Using all SBS, DBS and INDEL variants from protein-coding genes, we ran dNdScv[45] to find significantly mutated genes using all SBSs and INDELs variants from protein-coding genes. This model can test the normalized ratio of each non-synonymous mutation type individually (missense, nonsense, and splicing) over background (synonymous) mutations whilst correcting for sequence composition and mutational signatures. A global $q$ value ≤0.1 was used to identify statistically significant driver genes. In parallel, also the positive selected genes as observed in were surveyed in every sample.

### Power analysis

In this study we relied on a sample size set similar to other studies that also used healthy tissue derived organoids to investigate mutation accumulation. To further explore the sample size of the current study, we have conducted a power analysis. Details about the power analysis can be found in Supplementary Note 1.

### Statistics

Unless otherwise stated, we performed a Wilcoxon rank-sum test to compare continuous variables (for instance the absolute contribution of mutational signatures vs. treated and not treated). All statistical tests were one-sided (enrichment of treatment-induced mutations) and considered statistically significant when $p$ value < 0.01. R version 4.1.2 was used for the statistical analyses.

### Reporting summary

Further information on research design is available in the Nature Research Reporting Summary linked to this article.

## Data availability

Whole-genome sequencing data stored in BAM format of treated donors generated in this study have been deposited at the European Genome-phenome Archive (http://www.ebi.ac.uk/ega/) under accession number EGAS00001006042. Access can be obtained by UMCU DAC. Raw sequencing data of colorectal and liver ASCs from untreated donors were respectively downloaded from EGA download portal under accession numbers EGAS00001000881 and EGAS00001001682. We also included mutation data from healthy colon tissue (https://doi.org/10.1038/s41586-019-1672-7) downloaded from https://raw.githubusercontent.com/HLee-Six/colon_microbiopsies/master/signature_extraction/subsitutions_hdp_signature_extraction/sbs_category_counts.txt and mutation data from healthy liver tissue (https://doi.org/10.1038/s41586-019-1670-9) downloaded from Mendeley Data platform with the identifier (https://doi-org.proxy.library.uu.nl/10.17632/ktx7jp8sch.1). The processed mutations in this study are provided in the Source Data file. The mutation calls from the sequencing data used in this study are available in the Zenodo database under accession code (https://doi.org/10.5281/zenodo.7057493). Source data are provided with this paper.

## Code availability

All code and filtered vcf files from anti-cancer treated colorectal and liver ASCs are freely available at https://github.com/UMCUGenetics/treatment_muts_healthy_tissue/.

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

## Acknowledgements

We would like to thank USEQ from UMCU for sequencing the organoid lines. We thank Jan Hendrik Venhuizen, Jorieke Salij and Anneta Brousali of the Utrecht Platform for Organoid Technology for patient inclusion and for arranging the logistics of tissue acquisition. We are also particularly grateful to all the colorectal cancer patients hospitalized at UMCU for donating their tissue.

## Author contributions

E.K. performed wet-lab experiments. A.V.H. performed the bioinformatical analyses. E.K. and A.V.H. wrote/edited the paper. O.K. and E.C. edited the paper and provided discussion. E.K., A.V.H. and E.C. were involved in the conceptual design of the study. All authors proofread, made comments, and approved the paper.

## Competing interests

The authors declare no competing interests.
