## [Peer review file · Nature Communications]

REVIEWER COMMENTS

Reviewer #1 (Remarks to the Author): Expert in somatic mutational signature analysis and genomics

Kuijk et al., have studied the effect of CapOx (5-FU and Oxaliplatin) and/or radiotherapy on normal stem cells of colon and liver tissues from individuals with either colorectal and or metastatic liver cancers. Their results highlight the tissue-specific respond to chemotherapy treatment across non-cancerous cells. They suggest colon is more vulnerable to chemo treatments, especially platinum-based treatments (e.g., SBS35) compared to liver which seem to be more protected. This vulnerability in turn can increase the risk of secondary cancer incident in colonic tissues.

Some recent studies of somatic mutations in normal cells using laser capture microdissection (e.g., Moore, et. al., 2021 and Lee-Six et al., 2019) or single cell derived colonies (e.g., Pich et al., 2021) have highlighted the footprints of chemotherapy on mutational landscape and clonal dynamics of normal cells. However, more work in this area is certainly required to systematically measure the effect of chemo on normal cells and to assess the risk of further cancer transformation. Therefore, this work is certainly relevant and can be interesting to the readers of this journal. However, I have listed some comments below to be clarified by the authors:

- What is the average number of treatment cycles for liver metastases? Biopsies from CRC patients were collected around ~1.5 months after latest CapOx, while for liver metastases samples were biopsied ~15 months post treatment. How variation to the time of biopsy may explain some of the variation in effect of chemo they have reported between these two groups? The authors indeed have suggested that damaged cells may have been removed from the liver due to longer time from the treatment to the biopsy collection. However, liver stem cell turnover is slower compared to colonic crypts stem cells. Hence, I wonder other factors might be at play such as variation in clonal dynamics between the two tissues?
- It is quite surprising that the 58 and 76 yrs-old CapOx-treated donors as well as the 5-FU-only treated donor did not show any elevation in SBS burden. How the authors explain this observation?
- Is there any correlation between SBS35 load and duration of the treatment and or different sections of colon that were biopsied?
- They have shown that the effect of 5-FU is more heterogenous across the individuals studied. Similar observation was also made in AML cases by Pich et al., 2021. Does the dose of 5-FU vary between individuals and if so, is there any correlation between the dose and SBS17 burden?

- The observation of sub-clonal platinum-induced mutations is quite unexpected. Although the authors suggested alternative possibility for this observation, yet it highlights that some of the chemo effect might be missed with their method. Can the authors give an estimate of the fraction of chemo effect that might have missed due to their sub-clonality nature?
- “Our findings reveal that both CapOx chemotherapy and radiotherapy are mutagenic in colorectal ASCs and significantly increase the mutational burden in normal non-cancerous cells beyond typical age-related mutation accumulation.” It would be great if the authors can expand on this observation by providing more insights into the type of mutations with respect to their functional consequences.

Minor comments:

- Overall, the way the experimental design has described is slightly confusing. It is unclear to me how many colonies were sequenced per individual. A simple summary table to include this information as well as, type of treatment, dosage, length of treatment and the time interval from the last treatment to sample collection would be very helpful for the readers.
- Fig. 2 left and right panels are not labelled so I assumed the left panel is colon and right panel is liver. In Fig. 2f, the anti-correlation of indel with age even in healthy individuals is unexpected.

Reviewer #2 (Remarks to the Author): Expert in colorectal cancer organoids, therapy, and stem cells

The manuscript by Kuijk et al. identifies the mutational landscape in normal colonic and liver cells of CRC patients after being treated with several cycles of therapy (chemo and/or radiotherapy). This is an relevant study that addresses the key question of how normal tissue is damaged after treatment. The authors use whole genome sequencing to demonstrate that the mutational pattern in the liver does not change significantly after chemo but intestinal cells are affected (mainly by Oxaliplatin).

The work is original as the mutational landscape by NGS has not been well reported in normal intestinal and liver cells. It is an interesting study, with a good experimental set up. The group has an excellent

background in NGS/cancer genetics and in identifying mutational landscapes of different cell types. The results are well interpreted, however a more exhaustive report would need to be addressed.

Major points:

1. My main concern is the low number of CRC samples analysed and the poor description in the text about the samples/crypts that were sequenced. Are the results statistically significant? Can they do further analyses to be sure that the samples are representative? If they do power calculations/sample size, what would be the result? The liver data looks convincing but the authors would need to be sure that the rest shown is also representative.

2. Why the authors discuss the 5FU influence as it has an impact, when 5 out of 7 show no effect and another individual only in one of the samples?

3. Colorectal cancer histology/pathology is not shown/described however. which part of the colon the samples were resected from? it is important to know because may add more variability to the N analyzed (in addition to the age factor which is cover a big range). The text does not address it and this may have an impact. Did the authors see any difference in SBS and ID mutations by anatomical site in the colon and rectum? this may need to be discussed as a potential limitation.

4. In my opinion, the telomere section is roughly described and data are not included in the abstract/main figures. Makes sense to leave it? I would suggest to leave it for a better analysis.

5. The irradiation findings are not so novel as have already been described in similar approaches/cancers (e.g. 10.1038/ncomms12605). Authors refer to this in the introduction but do not discuss it later on.

6. The crypts were expanded from few cells, but as confirmed by the authors the results show that there could be more than one stem cell in each sample before expansion. Did they analyse this in depth? Using stem cell markers, flow cytometry and quantification? through-out the text is written continuously that they are working with single adult stem cells, is this always true? For example, figure legend states “single adult SCs solutions” , I think this can take the reader to confusion.

7. A similar set up was done by Halazonetis group (Switzerland) where single crypts were expanded into organoids from mouse samples (AKP) before sequencing. A previous manuscript from the same group using APC min also revealed the mutation change when crypts were cultured more than 4 months.

Authors should mention and compare results. The authors should also look at the AKP manuscript to do a scheme similar to show the number of crypts expanded/sequenced from each sample.

Minor points:

1. Graphs in figure 3 and 4 are too small. Supplementary data should also be shown with a bigger size.
2. The statistics section is poorly described. The normal and non-normal distribution should be better study than doing "assumptions" or "likely".
3. It should be written in the abstract how many individuals for each sample type were used to do the study.
4. Methods section: "Sorbitol and 0.5mM dithiothreitol (all purchased from Merck) until the supernatant was." (words are lacking here).

Reviewer #3 (Remarks to the Author): Expert in colorectal and hepatic organoids, stem cells

Kuijk et al. analyzed mutation patterns in chemotherapy-treated colon/liver tissue using organoid technology and estimated the number of mutations induced by chemotherapies. This is a fascinating study based on a valuable dataset from clinical specimens. The finding is novel in that chemotherapy-induced mutations in a tissue-specific manner. I would like to recommend its publication in Nature Communications, but several points are to be addressed as follows.

1. The "liver organoids" used in this study are actually derived from intrahepatic cholangiocytes and thus should be referred to as intrahepatic cholangiocyte organoids (ICO) (Marsee et al. Cell Stem Cell 2021).
2. As described in the discussion, the weak point of this paper is a limited number of specimens, and it is unclear why only a fraction of organoid clones showed the 5-FU signature. Are the two lines of

organoids that showed the 5-FU signature associated with radiation therapy (or rectum origin)? I think the authors should provide experimental evidence that

3. ICOs were devoid of platinum signatures. The authors provided an interesting explanation that liver ASCs have more effective inherent mechanism to protect against chemo-induced mutagenesis. However, there is an alternative explanation that all liver ASCs that gained platinum signature died out or underwent senescence in vivo (thus, they cannot be recovered as organoids). In addition, liver ASCs are non-cycling, and non-cycling cells may gain fewer mutations by platinum drugs? To exclude these possibilities, the authors should treat ICO with platinum and determine whether ICO can gain platinum signature when proliferating in vitro.

4. The authors excluded escaping from drugs in liver ASCs. However, oxaliplatin is a renal excretion drug. Why didn't they consider this possibility?

Minor points

1. Fig.1 is not very self-explanatory. I think it is better to label colons and cholangiocytes in the figure. The color codes for 5-FU+platinum+radiation and 5-FU+platinum are very similar, and I cannot distinguish the two colors. I cannot see which plots are significant in the figure.

Response to REVIEWER COMMENTS

Reviewer #1 (Remarks to the Author): Expert in somatic mutational signature analysis and genomics

Kuijk et al., have studied the effect of CapOx (5-FU and Oxaliplatin) and/or radiotherapy on normal stem cells of colon and liver tissues from individuals with either colorectal and or metastatic liver cancers. Their results highlight the tissue-specific respond to chemotherapy treatment across non-cancerous cells. They suggest colon is more vulnerable to chemo treatments, especially platinum-based treatments (e.g., SBS35) compared to liver which seem to be more protected. This vulnerability in turn can increase the risk of secondary cancer incident in colonic tissues.

Some recent studies of somatic mutations in normal cells using laser capture microdissection (e.g., Moore, et. al., 2021 and Lee-Six et al., 2019) or single cell derived colonies (e.g., Pich et al., 2021) have highlighted the footprints of chemotherapy on mutational landscape and clonal dynamics of normal cells. However, more work in this area is certainly required to systematically measure the effect of chemo on normal cells and to assess the risk of further cancer transformation. Therefore, this work is certainly relevant and can be interesting to the readers of this journal.

However, I have listed some comments below to be clarified by the authors:

- What is the average number of treatment cycles for liver metastases? Biopsies from CRC patients were collected around ~1.5 months after latest CapOx, while for liver metastases samples were biopsied ~15 months post treatment. How variation to the time of biopsy may explain some of the variation in effect of chemo they have reported between these two groups? The authors indeed have suggested that damaged cells may have been removed from the liver due to longer time from the treatment to the biopsy collection. However, liver stem cell turnover is slower compared to colonic crypts stem cells. Hence, I wonder other factors might be at play such as variation in clonal dynamics between the two tissues?

Clinical details are described in Suppl table 1 including the number of treatment cycles. We have also included this information in figure 1 panel A. The liver patients included in this study have undergone more CAPOX treatment cycli than the colon patients (CAPOX treatment cycli of 5.8 ± 3.5 for liver patients and 3.2 ± 1.7 for colon patients; mean \pm standard deviation [SD]) and thus the lack of treatment-related mutations in liver ASCs cannot be explained by a lower number of treatment cycles. We have now included this sentence in the main text.

On average, there is indeed more time between the end of CAPOX treatment regime and surgical removal of tumor and surrounding healthy tissue (= time of biopsy) for liver patient than colon patients. However, healthy tissue of 2 liver donors (Donor 8, 10) was collected 1 month after the end of CAPOX treatment, which is below the average of 1.5 month for colon donors. All 5 liver ASCs of these 2 liver donors showed a complete absence of CAPOX induced mutations as well, indicating that time between treatment and biopsy is likely not the cause for the lack of CAPOX related mutations for liver ASCs. However, we cannot formally exclude the possibility of a highly efficient clearance of treatment damaged healthy liver ASCs. We consider this hypothesis unlikely because 5-

FU or platinum therapies rarely induce liver damage¹. We have added this text to the discussion section of the manuscript.

Given the very slow liver stem cell turnover rate, we expect that healthy liver tissue harbour different clonal compositions than colon tissue. Sequencing a clonal group of nearby cells that have recently derived from a single (adult stem) cell (e.g. by micro laser biopsy sampling as conducted in Moore et al 2021) could shed more light on clonal dynamics of treatment-induced mutations. However, we used a single-cell based whole genome sequencing methodologies that integrate clonal expansion of a single cell and thus clonal dynamics cannot be studied with this approach.

Although it has been demonstrated that treatment changes the clonal architecture in cancer tissue due to selection of treatment driver resistance mechanisms (Martínez-Jiménez et al., 2022 BioRxiv), it may induce similar effects in healthy tissue. However, we have not found a different driver landscape between treated and untreated tissue in either liver or colon adult stem cells. Nevertheless, the number of samples we have sequenced here are not sufficient to be fully conclusive, but we also believe that studying clonal dynamics, although very interesting, would require different experimental approaches that are beyond the scope of this study.

- It is quite surprising that the 58 and 76 yrs-old CapOx-treated donors as well as the 5-FU-only treated donor did not show any elevation in SBS burden. How the authors explain this observation?

There is natural variation in the correlation between the age and the number of mutations, which increases with age. This means that the older the patients are, the more difficult it is to get significant effects of increases in mutational burden as illustrated by higher standard deviation at older age in Figure 2a and 2b. Thus, while we can clearly identify treatment-induced mutations in these donors by mutational signature analysis, the effect of the increase in mutations as a result of the treatment is not large enough to become apparent by mutational burden analysis alone.

We have clarified this by changing the results section as follows: “The SBS mutation burden of the 58 and 76 year old CapOx-treated donors, as well as the 5-FU-only treated donor, was within the natural variation in SBS mutation burden in the correlation with the age”

- Is there any correlation between SBS35 load and duration of the treatment and or different sections of colon that were biopsied?

There was no correlation between SBS35 mutation load and treatment duration. See comments on reviewer 2 related to colon anatomical site in the colon and rectum.

- They have shown that the effect of 5-FU is more heterogenous across the individuals studied. Similar observation was also made in AML cases by Pich et al., 2021. Does the dose of 5-FU vary between individuals and if so, is there any correlation between the dose and SBS17 burden?

Unfortunately, treatment dose information was not available for the current study. Nevertheless, only 5 of 28 colon ASCs showed 5-FU related mutations. Of these 5 colon ASCs, 4 (of 4) samples

came from the 24-year old donor and 1 (of 3) of the 66-year old donor. Thus, even if the 5-FU dose of each patient was shared, we would not observe any relationship between the given 5-FU dose and SBS-17 mutation burden because ~85% of the 5-FU exposed colon ASC show a complete absence of 5-FU related mutations. Moreover, internal 5-FU dose is also heavily influenced by pharmacogenetics as patients with (partial) DPD deficiency are less efficient in 5-FU clearance and typically display 5-FU toxicity during treatment (doi: 10.1016/j.ejca.2003.12.004). Finally, we observe some variability within the 24 year old patient, indicating that the variation in mutation accumulation is independent of 5-FU dosage.

- The observation of sub-clonal platinum-induced mutations is quite unexpected. Although the authors suggested alternative possibility for this observation, yet it highlights that some of the chemo effect might be missed with their method. Can the authors give an estimate of the fraction of chemo effect that might have missed due to their sub-clonality nature?

This information can be depicted from Suppl Figure 12 where we computed the % increase in treatment related mutations when considering all (clonal + subclonal) treatment mutations as compared to the clonal treatment mutations (VAF > 30%), which is equivalent to the fraction of missed 5-FU and platinum mutations to their subclonality. From the 25 oxaliplatin treated colon ASCs, 7 ASCs showed no increase in mutation load between all and clonal mutations (no missed subclonal treatment mutations), 16 showed an increase of ~50% in platinum mutation load and 2 samples showed an increase of 100% or more in both platinum SBS and DBS mutations.

We have adapted the main text as follows: "Nevertheless, only 2 ASCs (from the 24-year-old donor) showed a remarkable subclonal platinum mutation contributions of 100% or more and the inclusion of subclonal platinum mutations had a low impact (less than 50%) on the platinum mutational load for all other (n=25) ASCs (Supp. Fig. 12b,c)."

- "Our findings reveal that both CapOx chemotherapy and radiotherapy are mutagenic in colorectal ASCs and significantly increase the mutational burden in normal non-cancerous cells beyond typical age-related mutation accumulation." It would be great if the authors can expand on this observation by providing more insights into the type of mutations with respect to their functional consequences. This is indeed a very intriguing question. Although our approach is highly sensitive to detect the passenger mutational consequences of anticancer therapy, it is not optimal to identify potential drivers of secondary malignancies due to the low sample sizes and the relatively small proportion of coding sequence in the genome. Modelling functional impact and risks based on the limited number of coding mutations in our data set would result in highly unreliable risk estimates and thus incorrectly inform patients.

Minor comments:

- Overall, the way the experimental design has described is slightly confusing. It is unclear to me how many colonies were sequenced per individual. A simple summary table to include this information as well as, type of treatment, dosage, length of treatment and the time interval from the last treatment to sample collection would be very helpful for the readers.

Besides dosage, all this information is included in Suppl table 1 which we now have summarized in Panel A of main figure 1. We have also more comprehensively explained the experimental set up in the main text.

- Fig. 2 left and right panels are not labelled so I assumed the left panel is colon and right panel is liver.

Thank you for noticing. We have adapted the figure accordingly.

In Fig. 2f, the anti-correlation of indel with age even in healthy individuals is unexpected.

In general, there is no strong correlation between INDEL load and age in the liver. For example, the healthy (and diseased) liver tissue study from Brunner et al 2019 shows a slightly higher SBS mutation load in the older healthy liver samples while the indel mutation load remains constant. Also in our current study, this correlation is weak/not significant and we therefore caution not to overinterpret the figure based on visual inspection.

Reviewer #2 (Remarks to the Author): Expert in colorectal cancer organoids, therapy, and stem cells

The manuscript by Kuijk et al. identifies the mutational landscape in normal colonic and liver cells of CRC patients after being treated with several cycles of therapy (chemo and/or radiotherapy). This is an relevant study that addresses the key question of how normal tissue is damaged after treatment. The authors use whole genome sequencing to demonstrate that the mutational pattern in the liver does not change significantly after chemo but intestinal cells are affected (mainly by Oxaliplatin).

The work is original as the mutational landscape by NGS has not been well reported in normal intestinal and liver cells. It is an interesting study, with a good experimental set up. The group has an excellent background in NGS/cancer genetics and in identifying mutational landscapes of different cell types. The results are well interpreted, however a more exhaustive report would need to be addressed.

Major points:

1. My main concern is the low number of CRC samples analysed and the poor description in the text about the samples/crypts that were sequenced.

We have sequenced and included an additional set of 16 samples (13 from colon cohort and 3 from liver cohort) leading to a final set of 42 ASC samples. Overall, findings remain unchanged except that mutational signature analysis does not detect platinum related mutation in 2 additional samples of the 76-year old colon donor (n=3). This may be explained by the low number of CAPOX treatment cycles because this patient has undergone only a single treatment cycle. Therefore, the number of platinum-induced mutations is under the detection limit of mutation signature analysis. The information on the samples sequenced can be found in Suppl table 1 and as a table in Panel A of revised figure 1.

Are the results statistically significant? Can they do further analyses to be sure that the samples are representative? If they do power calculations/sample size, what would be the result? The liver data looks convincing but the authors would need to be sure that the rest shown is also representative.

In this study we relied on a sample size set similar to other studies that also used healthy tissue derived organoids to investigate mutation accumulation (Jager et al 2019, Drost et al 2017, Kuijk et al 2020, Nguyen et al, 2022). To further explore the sample size of the current study, we have conducted a power analysis as in Nguyen et al, 2022. For this, we first measured the Cohen's D to compute the standardized mean difference of the mutation load between platinum treated and untreated donors. Using the `cohen.D` function from `effsize` R package, we found a normalized effect size of 1.51. A similar normalized effect size (Cohen's D = 1.52) was observed when we assessed the excess in mutation load (observed mut load - predicted mut load using linear mixed model regression analysis) between platinum treated and untreated donors.

To assess the reliability of measured Cohen's D scores on mutation load given our number of samples, we performed a power analysis using the `pwr.t2n.test()` function of the `pwr` R package to calculate hypothetical detectable effect sizes using a significance level of 0.05 and a power of 0.8. It is generally accepted that power should be approximately 80%. Given that we have 5 untreated patients and 6 treated patients, we expect to be able to detect an effect size (Cohen's D) of 1.6 which is approximately equal to the computed Cohen's D values on the mutation load and excess mutation load. Thus, the sample sizes in our study would allow us to detect effect sizes as hypothetical detectable effect sizes with a power score of 80%.

Similarly, we applied this approach on the platinum and 5-FU mutation contributions. Using the SBS35 and SBS17 mutation contributions, we obtained Cohen's D scores of 2.7 (large) and 0.7 (medium) for platinum and 5-FU mutation contributions, respectively, between the treated and untreated donor group. The Cohen's D score on the platinum DBS (DBS5) contribution was 2.5, which is similar as the SBS35 mutation contribution. Thus, the normalized effect size for platinum mutation load is well beyond the hypothetical detectable effect size given the number of samples (1.6), but not for 5-FU mutation contribution. This is in line with our observations that 5-FU mutations were only detected in few samples of 2 (of 7) donors.

Lastly, we found that the Cohen's D score for radiation induced indels (ID8) between untreated and radiation treated donors was also higher (ie 1.8) than the hypothetical detectable effect size with higher power.

We have included this in Methods section and all details in a Suppl Note:

Power Analysis

In this study we relied on a sample size set similar to other studies that also used healthy tissue derived organoids to investigate mutation accumulation. To further explore the sample size of the

current study, we have conducted a power analysis as in Nguyen et al, 2022. Details about the power analysis can be found in Suppl Note 1.

2. Why the authors discuss the 5FU influence as it has an impact, when 5 out of 7 show no effect and another individual only in one of the samples?

The specificity of the 5FU mutational signature allows us to very accurately determine the presence of 5FU-induced mutations. As we have demonstrated previously, this enables the detection of 5FU induced mutations in whole genome sequencing data of 5FU treated colorectal cancer patients, albeit not in every patient, the reason for which is unknown (Christensen et al). In our current study, most of the cells did not show 5FU induced mutations. However, based on the highly characteristic mutational footprint of 5FU, we are very confident that 18% of the colon cells harbor 5-FU induced mutations. We prefer not to trivialise this effect, because in some cells 5FU treatment equals ~ 10 years of mutations caused by normal age-related processes. Therefore, we can conclude that 5FU can have a mutational impact on healthy cells and may cause secondary malignancies later in life. Fortunately for the patients, most healthy cells escape 5FU induced mutagenesis, which is also an important message. We have modified the abstract to address that in most of the colon ASCs we did not observe 5-FU induced mutations: “The effects of 5-FU treatment were more variable between cells and patients, ranging from a complete absence of 5-FU mutagenesis in >80% of the individual colon ASCs to the accumulation of up to 500 mutations in the remaining individual colon ASCs.”

3. Colorectal cancer histology/pathology is not shown/described however. which part of the colon the samples were resected from? it is important to know because may add more variability to the N analyzed (in addition to the age factor which is cover a big range). The text does not address it and this may have an impact. Did the authors see any difference in SBS and ID mutations by anatomical site in the colon and rectum? this may need to be discussed as a potential limitation.

Clinical info on tumor type and sublocation is included in Suppl table 1 and in panel A of figure 1. We have changed the column names accordingly in order to better interpret the clinical details.

The study from Lee-Six have shown that age-related mutations vary between anatomical location of the colon tissue, because of a slightly lower proliferation rate at the end of the colon (~12 SBS1 mutations per year) compared to the right colon section (~18 SBS1 mutations per year). Because all our samples are from the descending/sigmoid/rectal parts of the colon and not from the ascending/transverse sections, the effect of location on variation is minimal. Moreover, any putative location-dependent variation cannot explain the complete absence of 5-FU mutations in 85% of the colon ASCs. Because, our results demonstrate that chemotherapy mutation accumulation varies between tissues, this may also be true for the ascending and transverse sections of the colon, which were not examined in the present study. We have therefore included this possibility in the discussion section of our revised manuscript: “ future work should aim to further elucidate the mutagenic effects of chemotherapies on other organ systems and organ anatomical sublocations “

4. In my opinion, the telomere section is roughly described and data are not included in the abstract/main figures. Makes sense to leave it? I would suggest to leave it for a better analysis.

We agree with the reviewer and we removed this section in the revised version of our manuscript.

5. The irradiation findings are not so novel as have already been described in similar approaches/cancers (e.g. 10.1038/ncomms12605). Authors refer to this in the introduction but do not discuss it later on.

The study from Behjati et al., is indeed the first study to assess radiation-induced mutagenesis in which they reported an increase in mutation contribution of an indel signature (sized between 1–100 base pairs) and a balanced inversion signature. We have referred to this publication in radiation section as follows:

This signature was mostly similar to COSMIC ID-8 (cos sim =0.66). ID-8 mutations were previously found to be enriched in human cancers following radiation therapy^{2,3}.

In addition to small deletions, we investigated SVs because radiation-treated tumors show radiotherapy-associated increases of inversions and large-scale deletions^{2,3}.

As with platinum and 5-FU mutational signatures, we do not claim that these radiotherapy-related signatures are novel, but we use these signatures as barcode to also quantify the mutagenic impact of radiotherapy in healthy rectum adult stem cells, which has not been described before.

6. The crypts were expanded from few cells, but as confirmed by the authors the results show that there could be more than one stem cell in each sample before expansion. Did they analyse this in depth? Using stem cell markers, flow cytometry and quantification? through-out the text is written continuously that they are working with single adult stem cells, is this always true? For example, figure legend states “single adult SCs solutions”, I think this can take the reader to confusion.

Most of the treatment induced neutral passenger mutations are only present in one or a few cells. Therefore, sequencing and characterizing the genome-wide in vivo mutations of a single cell enables accurate quantification of the mutational contribution from late active mutation processes just before tissue sampling. Recently, we optimized and presented an alternative method for cataloging mutations in individual human ASCs without the necessity of using error-prone whole-genome amplification [5]. Here, tissue derived single ASCs are expanded in vitro into clonal organoid cultures to generate sufficient DNA for accurate WGS. The culture conditions have been optimized to promote the growth of intestinal stem cells over other cell types such as differentiated cells or stromal cells. This method is therefore particularly well suited to study and quantify treatment-induced mutations in ASCs in vivo. However, performing a clonal step is highly challenging with colon organoids that are not readily expanded from single cells, particularly from the relatively small biopsies used for the present study. Therefore, we adapted the protocol to maximize the chance of clonal expansion, by first establishing the culture followed by severe fragmentation and serial dilutions at the first split, where individual tiny fragments are most likely derived from individual cells at isolation or have a very recent common ancestor in vivo. As is evident from our observation that we can detect treatment induced mutations, this method is adequate for capturing very recent mutations. However, in rare cases, the small cell clump that gave rise to the clonal culture may

represent more than one stem cell of the crypt at the time of isolation. As experimental evidence of the success of the clonal step we examined the VAF peaks. The clonal step worked as expected in all of the lines as illustrated by a highly characteristic VAF-peak at 50% across all samples. Lower VAF peaks indicate that mutations are not shared by the single stem cell ancestor and are typically obtained by laser-capture microdissection of crypts (See for example Suppl Fig 1d,e in Lee-six et al., 2019).

The VAF-peaks at 50%, and the strong decrease of mutations at VAF at 30%, indicate that we have assessed somatic mutations of single adult stem cells (see also protocol guidelines in Jager et al., 2017⁴). Moreover, the absence of subclonal treatment-induced mutations in ~85% of colon ASC samples confirms the clonal outgrowth of a single ASCs. However, 2 out of 25 platinum treated colon samples showed subclonal platinum DBS and SBS mutations with similar contribution as clonal platinum DBS and SBS mutations. This indicates a fairly recent single stem cell ancestor at the beginning of the treatment but diverged into 2 (or more) stem cells during the course of the treatment. In these two clones we cannot exclude the possibility that these have been derived from two or more cells resulting in an underestimation of the real mutational impact of the anticancer therapy.

We have adapted the introduction and methods section of the manuscript to better explain the technical details of the protocol and included the experimental VAF analysis in the results.

7. A similar set up was done by Halazonetis group (Switzerland) where single crypts were expanded into organoids from mouse samples (AKP) before sequencing. A previous manuscript from the same group using APC min also revealed the mutation change when crypts were cultured more than 4 months. Authors should mention and compare results. The authors should also look at the AKP manuscript to do a scheme similar to show the number of crypts expanded/sequenced from each sample.

The mutational impact of culturing in human adult stem cells from colon and liver tissue have been well-documented in a previous study by our group⁵. In that study we have shown that in vitro culturing induces SBS-18 mutations which are related to oxidative stress, and that low-oxygen culturing conditions decrease SBS-18 mutation rate during culturing. We minimized the time in culture 7-10 days before we performed the clonal step, which is far below the 4 month period of the study by the Halazonetis group. The potential small contribution of the derivation/culturing before the clonal step was applied is negligible (see also validation in our in vivo study⁶)). Moreover, culturing induced mutations have a very different mutation context than the highly specific 5-FU and platinum-induced mutations (see also Christensen et al where we identified a clear 5FU signature apart from the in vitro signature by the in vitro treatment of intestinal organoids). Thus, even the few in vitro induced mutations that end up in the mutation matrices from clonal mutations (representing the in vivo induced mutations) will have no impact on the quantitative assessment of treatment induced mutations.

Finally, any in vitro induced mutations after the clonal step are subclonal after sequencing the bulk isogenic organoid culture and are discarded in downstream analysis .

Minor points:

1. Graphs in figure 3 and 4 are too small. Supplementary data should also be shown with a bigger size.

Main figure 3 and 4 as well as Suppl figures have been adapted.

2. The statistics section is poorly described. The normal and non-normal distribution should be better study than doing "assumptions" or "likely".

Thank you for pointing this. We have revised the statistical section.

3. It should be written in the abstract how many individuals for each sample type were used to do the study.

We have included the mean number of sequenced ASC samples per donor in the abstract. A complete overview on sample size per donor is now also provided in main figure 1 and in supplementary table 1.

4. Methods section: "Sorbitol and 0.5mM dithiothreitol (all purchased from Merck) until the supernatant was." (words are lacking here).

Thank you for pointing this out. We have completed this particular sentence ("...until the supernatant was clear.")

Reviewer #3 (Remarks to the Author): Expert in colorectal and hepatic organoids, stem cells

Kuijk et al. analyzed mutation patterns in chemotherapy-treated colon/liver tissue using organoid technology and estimated the number of mutations induced by chemotherapies. This is a fascinating study based on a valuable dataset from clinical specimens. The finding is novel in that chemotherapy-induced mutations in a tissue-specific manner. I would like to recommend its publication in Nature Communications, but several points are to be addressed as follows.

1. The "liver organoids" used in this study are actually derived from intrahepatic cholangiocytes and thus should be referred to as intrahepatic cholangiocyte organoids (ICO) (Marsee et al. Cell Stem Cell 2021).

Because colon stem cells and intrahepatic cholangiocyte organoids are both adult stem cells (ASCs) we use this latter terminology. To further specify the type of liver ASCs we use the suggested terminology and refer to the suggested paper at the end of the introduction: "We studied mutations in ASCs derived from the slowly renewing liver (cultured as intrahepatic cholangiocyte organoids⁷)."

2. As described in the discussion, the weak point of this paper is a limited number of specimens

To overcome this limitation, we have sequenced and analyzed extra samples and performed a power analysis. See also our answers to the comments by reviewer 2.

3. and it is unclear why only a fraction of organoid clones showed the 5-FU signature. Are the two lines of organoids that showed the 5-FU signature associated with radiation therapy (or rectum origin)?

Of the 5 colon ASCs that show a contribution of the 5-FU mutational signature, 4 (of 4) samples came from the 24-year old donor and 1 (of 3) of the 66-year old donor. Both donors were diagnosed with rectum and thus treated with radiation, opening the possibility that 5-FU signature could be associated with these covariates. However, 5-FU related mutations have been documented in colon tissue in patients not being treated with radiation (<https://doi.org/10.1101/2021.04.14.437578>; and vice versa, no enrichment of 5-FU related mutations (SBS17a/b) have been described in radiation treated cancers^{2,3}. These results indicate that 5-FU and radiation mutation processes operate independently.

I think the authors should provide experimental evidence that ICOs were devoid of platinum signatures.

Platinum leads to interstrand crosslinks and thereby causes highly characteristic DBS mutations as has been experimentally demonstrated in platinum-treated isogenic pluripotent stem cells⁸. Double base mutations are highly informative because they include two neighbouring SBS mutations located on the same strand. These platinum related mutation scars are thus highly robust to assess the mutational impact of platinum. In Suppl Figure 8, we show that all sequenced liver ASCs (ICOs) don't show any increase of CT>AT and CT>AA DBS mutations compared to untreated liver ASCs. In sharp contrast, nearly all colorectal ASCs from patients treated with platinum show these DBSs (Fig. 3d-e).

The authors provided an interesting explanation that liver ASCs have more effective inherent mechanism to protect against chemo-induced mutagenesis. However, there is an alternative explanation that all liver ASCs that gained platinum signature died out or underwent senescence in vivo (thus, they cannot be recovered as organoids).

This hypothesis was also included in the results section : "Damaged cells may have been effectively cleared from the liver because the time between treatment and collection was mostly longer for the liver than for the colon, even though also no mutations were observed in the 2 liver samples that were collected 1 month after treatment." See also comment 1 from reviewer 1.

In addition, liver ASCs are non-cycling, and non-cycling cells may gain fewer mutations by platinum drugs? To exclude these possibilities, the authors should treat ICO with platinum and determine whether ICO can gain platinum signature when proliferating in vitro.

We agree with the reviewer that more focused in vitro validation experiments must be conducted to dissect the mechanisms of chemotherapy induced mutation accumulation and the role of proliferation in this process. However, these mechanistic studies are beyond the scope of the

current manuscript that is focused on the in vivo mutational impact of anticancer therapies in different organ systems. We have therefore decided to discuss our results in the light of previous studies (Oriol et al., 2021 Nature communications; Bertrums et al., 2022 Cancer Discover). Together, these findings indicate that platinum mutation effect is independent of cell proliferation.

We have included the following sentence to the discussion part: "Similar observations in mutational effect between platinum and nucleobase analogue (5-FU, thiopurines) drugs has been reported in treated hematopoietic cells^{9,10} and well-controlled in vitro validation experiments will dissect the mechanisms of chemotherapy-induced mutation accumulation and the role of proliferation in this process"

4. The authors excluded escaping from drugs in liver ASCs. However, oxaliplatin is a renal excretion drug. Why didn't they consider this possibility?

A highly efficient mechanism clearance of (oxali)platin based drugs from the vascular system (via renal excretion) could explain the absence of platinum mutations in liver ASCs, but it is not in line with the apparent and robust mutation effect in colon ASCs. Moreover, after uptake in the bloodstream, drugs and nutrients will travel through the portal vein into the liver, thereby exposing all liver cells to the platinum before it can be renally excreted.

Minor points

1. Fig.1 is not very self-explanatory. I think it is better to label colons and cholangiocytes in the figure. The color codes for 5-FU+platinum+radiation and 5-FU+platinum are very similar, and I cannot distinguish the two colors. I cannot see which plots are significant in the figure.

We have changed Fig 1 accordingly.

1. Andrade, R. J. *et al.* Drug-induced liver injury. *Nat Rev Dis Primers* **5**, 58 (2019).
2. Kocakavuk, E. *et al.* Radiotherapy is associated with a deletion signature that contributes to poor outcomes in patients with cancer. *Nat. Genet.* **53**, 1088–1096 (2021).
3. Behjati, S. *et al.* Mutational signatures of ionizing radiation in second malignancies. *Nat. Commun.* **7**, 12605 (2016).
4. Jager, M. *et al.* Measuring mutation accumulation in single human adult stem cells by whole-genome sequencing of organoid cultures. *Nat. Protoc.* **13**, 59–78 (2018).
5. Kuijk, E. *et al.* The mutational impact of culturing human pluripotent and adult stem cells. *Nat.*

- Commun.* **11**, 2493 (2020).
6. Blokzijl, F. *et al.* Tissue-specific mutation accumulation in human adult stem cells during life. *Nature* **538**, 260–264 (2016).
 7. Marsee, A. *et al.* Building consensus on definition and nomenclature of hepatic, pancreatic, and biliary organoids. *Cell Stem Cell* **28**, 816–832 (2021).
 8. Kucab, J. E. *et al.* A Compendium of Mutational Signatures of Environmental Agents. *Cell* **177**, 821–836.e16 (2019).
 9. Bertrums, E. J. M. *et al.* Elevated mutational age in blood of children treated for cancer contributes to therapy-related myeloid neoplasms. *Cancer Discov.* (2022) doi:10.1158/2159-8290.CD-22-0120.
 10. Pich, O. *et al.* The evolution of hematopoietic cells under cancer therapy. *Nat. Commun.* **12**, 4803 (2021).

REVIEWERS' COMMENTS

Reviewer #1 (Remarks to the Author):

The authors have address my questions and comments adequately. I have no further comments and I am happy with the revised version of the manuscript to go ahead for publication.

Reviewer #2 (Remarks to the Author):

I would like to thank the authors for their rebuttal letter. They have improved substantially the data and have discussed well their findings. They have also well discussed the limitations of their work.

Reviewer #3 (Remarks to the Author):

The labeling of "Liver ASC" is misleading for readers. Unfortunately, the authors were not responsive to my request and merely added a sentence in the introduction. I am afraid most readers would misunderstand that "hepatocytes" escape from mutations.

REVIEWERS' COMMENTS

Reviewer #1 (Remarks to the Author):

The authors have address my questions and comments adequately. I have no further comments and I am happy with the revised version of the manuscript to go ahead for publication.

We thank the reviewer for their appreciation of our responses and helpful comments.

Reviewer #2 (Remarks to the Author):

I would like to thank the authors for their rebuttal letter. They have improved substantially the data and have discussed well their findings. They have also well discussed the limitations of their work.

We thank the reviewer for their support and helpful comments throughout the review process.

Reviewer #3 (Remarks to the Author):

The labeling of "Liver ASC" is misleading for readers. Unfortunately, the authors were not responsive to my request and merely added a sentence in the introduction. I am afraid most readers would misunderstand that "hepatocytes" escape from mutations.

In the introduction we clearly state that we culture liver stem cells as intrahepatic cholangiocyte organoids. cholangiocytes are a distinct liver cell type than the hepatocytes in the liver and therefore we cannot conclude that "hepatocytes" escape from mutations.